# Skeletal Muscle ^31^P Magnetic Resonance Spectroscopy Study of Patients with Parkinson’s Disease: Energy Metabolism and Exercise Performance

**DOI:** 10.3390/diagnostics15202573

**Published:** 2025-10-13

**Authors:** Jimin Ren, Neha Patel, Talon Johnson, Ross Querry, Staci Shearin

**Affiliations:** 1Advanced Imaging Research Center, University of Texas Southwestern Medical Center, Dallas, TX 75390, USA; 2Department of Radiology, University of Texas Southwestern Medical Center, Dallas, TX 75390, USA; 3Department of Physical Therapy, University of Texas Southwestern Medical Center, Dallas, TX 75390, USA

**Keywords:** Parkinson’s disease, skeletal muscle, motor control, ATP synthesis, mitochondrial dysfunction, metabolism, exercise, plantar flexion, magnetization transfer

## Abstract

**Background/Objectives:** Parkinson’s disease (PD) is a progressive neurodegenerative disorder characterized by motor symptoms such as tremor, bradykinesia, rigidity, and postural instability. In the absence of disease-modifying therapies, exercise remains one of the few interventions shown to effectively reduce fall risk and improve mobility. However, it remains unclear whether skeletal muscle ATP metabolism is impaired in PD, and whether the benefits of exercise arise primarily from improvements in central motor control or peripheral metabolic adaptations. **Methods:** Fourteen individuals with PD and five healthy controls underwent kinetic ^31^P Magnetic Resonance Spectroscopy (MRS) to assess resting muscle ATP synthesis and dynamic ^31^P MRS during in-magnet exercise to evaluate oxidative phosphorylation in active muscle. **Results:** At rest, ATP synthesis rates mediated by ATPase and creatine kinase (CK) were on average 46 ± 23% and 24 ± 9% lower, respectively, in the PD group compared to controls (*p* < 0.005), suggesting peripheral mitochondrial dysfunction. During plantar flexion exercise at 15% of lean body mass, range of motion (ROM) was reduced by 22 ± 5% in PD participants (*p* = 0.01). Despite this, post-exercise recovery of phosphocreatine (PCr) and inorganic phosphate (Pi) was similar between groups. Recovery time constants for PCr and Pi correlated with participants’ total weekly exercise time, indicating a metabolic adaptation to regular physical activity. Modest ROM improvements were observed in both groups following calf-raise exercise training. **Conclusions:** Reduced skeletal muscle ATP metabolism may contribute to peripheral weakness in PD. Regular exercise appears to promote adaptive metabolic responses, highlighting the need for therapeutic strategies targeting both central and peripheral components of PD.

## 1. Introduction

Parkinson’s disease (PD) is the second most common neurodegenerative disorder in the United States, currently affecting nearly 1 million people with a continuously rising prevalence [1]. While PD is classically characterized by its impact on the basal ganglia and progressive degeneration of motor control, one critical factor to consider is weakness—the state of lacking strength, power, or effectiveness in performing physical tasks in daily life, a possible progressive motor manifestation secondary to the core motor symptoms including bradykinesia, rigidity, tremor, and postural instability [2]. Weakness contributes to morbidity, reduced functional mobility [2], increased risk of falls [3,4], and gait abnormalities [5], significantly diminishing the quality of life (QoL) in people with PD (PwP).

Although weakness is commonly attributed to impaired central motor control, as supported by the improved perception of weakness and motor performance following dopaminergic therapy, especially in tasks limited by bradykinesia and rigidity [6], its underlying mechanisms remain incompletely understood. Non-motor symptoms such as depression and apathy may affect the perceived weakness, potentially through reduced motivation and mental effort. Additionally, skeletal muscle weakness in PD may result from impaired energy metabolism due to mitochondrial maladaptation to chronic muscle disuse or underuse [7,8]. This may create a vicious cycle of weakness, inactivity, energy deficit, and progressive decline of motor control, an issue of particular concern given the absence of disease-modifying therapies to halt PD progression [9]. Recognizing this multifactorial interplay highlights the importance of quantitative assessment of skeletal muscle metabolism to inform intervention approaches, particularly those targeting exercise adaptation, which is the primary strategy with potential to slow functional decline [3,4,6,10].

Peripheral weakness plays a critical role in mobility impairment [11], especially in muscles of the lower extremities such as the gastrocnemius muscle (GM), which generates the greatest torque during walking and contributes to propulsion, knee stability, and balance during functional tasks [12]. Research from our group has shown that GM strength declines with increasing PD severity [5] and serves as a strong predictor of both gait speed and endurance [13]. Impairments in GM function are closely linked to reduced step length and limited plantarflexor excursion during walking [14,15], both of which are key features of altered gait kinematics in PD. Moreover, diminished GM strength contributes to higher fall rates in older adults [16], a major concern given the associations between falls, decreased walking speed, poorer QoL, and increased mortality in PD [17].

To better elucidate the metabolic basis of gastrocnemius (GM) muscle weakness in PD, this study will combine kinetic ^31^P magnetic resonance spectroscopy (MRS) to evaluate resting muscle ATP synthesis and dynamic ^31^P MRS to assess oxidative phosphorylation during exercise, an approach not previously applied in this population. Resting ATP synthesis will be quantified using the magnetization transfer method mEBIT (multi-modular exchange kinetics by band inversion transfer) previously developed by our team, which enables the differentiation and quantification of ATP flux through two major pathways: ATPase-mediated (de novo) synthesis and creatine kinase (CK)-mediated ATP recycling [18]. Additionally, the muscle oxidative phosphorylation function will be assessed by the dynamic change in the ^31^P MRS signal intensity of phosphocreatine (PCr) in exercised muscle. During exercise, PCr is depleted in response to muscle contraction and subsequently replenished during post-exercise recovery. The rate of post-exercise PCr recovery has been widely used for decades as a non-invasive spectroscopic marker of mitochondrial oxidative capacity. Together, these metabolic measurements may provide valuable insights into the underlying mechanisms of weakness in PD and inform more effective management strategies.

The literature examining resting and dynamic muscle characteristics in individuals with PD remains limited. Compared to the classic saturation transfer approach to explore resting muscle mitochondrial function, the current wideband inversion-recovery method has the advantage of generating pure magnetization transfer effects without off-resonance artifacts and is insensitive to frequency deviation [18]. Early work suggested that mitochondrial dynamics may contribute to PD pathogenesis, highlighting neuronal mitochondrial function as a potential source of therapeutic targets [19,20]. Most research in this field, however, has focused on mitochondrial dysfunction in the substantia nigra, with relatively few studies investigating its role in peripheral skeletal muscle [21,22]. Emerging evidence indicates that peripheral muscle mitochondrial impairments are also present in PwP [21,23]. For example, one study demonstrated significant reductions in mitochondrial ATP production and resting metabolite levels in the tibialis anterior of individuals with PD, suggesting that skeletal muscle mitochondrial function may provide valuable mechanistic and clinical insights [21]. In addition, Krumpolec et al. reported that aerobic-strengthening exercise improved peripheral muscle metabolism, measured using ^31^P MRS, and these improvements were associated with reductions in bradykinesia [23]. Building on this emerging body of work, the present study examined both resting and dynamic muscle metabolism in PwP.

## 2. Materials and Methods

### 2.1. Human Subjects

The protocol was approved by the Institutional Review Board at the University of Texas Southwestern Medical Center. Prior to the MR spectroscopy (MRS) study, written informed consent was obtained from all participants. The study involved 14 individuals clinically diagnosed with PD (age 66.7 ± 7.1 years, BMI 25.4 ± 4.8 kg/m^2^ and time since diagnosis 6.1 ± 4.2 years) who met the inclusion criteria of Hoehn and Yahr stages 1–4 (averaged 2.07 ± 0.62), as well as 5 age-matched healthy controls (age: 69.2 ± 6.5 years and BMI 27.3 ± 9.9 kg/m^2^). Participants were instructed to avoid any moderate or intense physical activity for 24 h before the study and to remain at rest for 20 min prior to the scan. All subjects tolerated the MRI scan and in-magnet exercise without issues.

### 2.2. MRS Protocol

#### 2.2.1. MRI Scanner, RF Coil, Exercise Device, and Subject Positioning

All participants were positioned head-first and supine in the MRI scanner (7T Achieva; Philips Healthcare, Best, The Netherlands), with the calf muscle of the non-dominant leg (for control participants) and the calf muscle of the leg with the most PD-related symptoms (for participants with PD) centered within the RF coil (Philips Healthcare). The coil was a double-tuned ^1^H/^31^P quadrature transmit-receive (TR) coil, comprising two tilted, partially overlapping 10 cm loops mounted within a half-cylinder-shaped plastic housing designed to fit the shape of the calf muscle. This coil was mounted on a solid base board of an ergometer that could be firmly attached to the MRI scanner table. Leg position was secured using Velcro straps, with a thick cushioning pad placed under the straps for added comfort. Participants were instructed to remain still and avoid muscle contractions, except during the supervised in-magnet exercise. 

The ergometer was a custom-built, 7T-compatible single-legged exercise device equipped with a pedal-driven pulley system, allowing participants to perform exercises at a prescribed workload. To ensure proper isolation of the exercised muscle and maintain stability, cushioned straps were applied around the ankle and knee of the exercising leg, with an additional strap placed around the pelvis to enhance body stability.

Each participant was scanned twice—once before and once after a 4-week progressive strengthening program—following the same scanning protocol. The exercise training program was designed to incorporate high-volume exercise to target the unique properties of the gastrocnemius muscle (the first two weeks involved 2 sets of heel raises to fatigue 3 times per week and then progressed to 3 sets, 3 times a week for the second two weeks, with each session lasting approximately 15 min utilizing progressive body weight loading). For imaging, axial, coronal, and sagittal T2-weighted turbo spin echo images were acquired for shimming purposes. Second-order, ^1^H-based automatic volume shimming was applied before the ^31^P spectral acquisitions.

#### 2.2.2. Dynamic ^31^P MRS with Plantar Flexion

To measure cellular oxidative phosphorylation in skeletal muscle, real-time dynamic ^31^P MR spectra were acquired from the exercising muscle. The exercise protocol consisted of 2 min of single-legged rhythmic plantar flexion, with the knee kept straight, performed in the supine position. During this exercise, the participant’s foot pressed against a pedal with a constant workload, personalized to 15% of the subject’s lean body mass (LBM). The pedal rotation had a hard stop at an angle of 30° at the full range of motion (ROM). A pulley system converted the in-magnet supine plantarflexion rotation into a vertical movement, lifting the prescribed resistance outside the scanner, enabling the actual ROM to be measured in distance.

The in-magnet exercise consisted of 60 repetitions, each lasting 2 s—1 s of active plantarflexion involving concentric contraction against the 15% LBM load, followed by 1 s of eccentric contraction under the same load to return to the initial position. Participants followed this exercise cadence given by the MRI operator through a telecom speaker, while a staff member in the scanner room monitored protocol compliance and recorded the ROM achieved by each participant during the exercise. The LBM value for each participant was calculated using the Boer formula (Women: 0.252 × weight [kg] − 148 × (weight [kg]/height [cm])^2^, and Men: 1.1 × weight [kg] − 128 × (weight [kg]/height [cm])^2^). The dynamic ^31^P MR spectra were acquired with a temporal resolution of 2 s, synchronized with the rate of the plantar flexion exercise. There was a total of 200 dynamic scans, including 15 at rest, 60 during exercise, and 125 during post-exercise recovery. The steady-state signals obtained from resting muscle served as an intensity reference, allowing for the assessment of signal dynamics during muscle contraction and subsequent recovery. All dynamic ^31^P spectra were aligned to the PCr signal (set to 0 ppm) to correct potential Bo variation due to the intrinsic and exercise-induced field inhomogeneity. To maintain data consistency and comparability, no Bo re-shimming was performed during or after exercise or between scans within the entire scan session.

#### 2.2.3. Kinetic ^31^P MRS at Resting State

ATP synthesis in resting skeletal muscle was measured using mEBIT (multi-modular exchange kinetics by band inversion transfer), a magnetization-transfer (MT) technique that generates MT effects by varying inversion-recovery time (TI) with tailored inversion modules, as previously described [18]. Three inversion modules were used: Module I generated MT effects at Pi and PCr by band inversion of all three ATP signals; Module II generated MT effects at Pi by band inversion of PCr and ATP; and Module III generated MT effects at ATP resonances by band inversion of Pi and PCr.

For all kinetic ^31^P measurements, a constant TR of 7 s was used, with 8 different TI acquisitions spaced logarithmically. Each TI acquisition was averaged over 8 consecutive scans and preceded by 1 dummy scan. The first TI point was 30 ms for all modules, while the last TI was 5 s for Module I, 6 s for Module II, and 3 s for Module III. The data acquisition time for each module was approximately 8 min. The inversion pulse used was a short, trapezoidal adiabatic pulse (pulse width = 38 ms, including 7 ms of pre- and post-ramp time) with a maximal B1 of 13 µT.

In addition to the mEBIT datasets with varying TI, a reference scan was conducted using the same TR of 7 s but without inversion, serving as a reference to assess MT effects. A fully relaxed ^31^P spectrum was also acquired at a long TR of 30 s to measure the thermodynamic equilibrium z-magnetization for converting forward and reverse kinetic rate constants between Pi, PCr, and ATP spins [18].

Other standard ^31^P MRS parameters included 4k sampling points, zero-filled to 8k before Fourier transformation, and a transmitter carrier frequency offset of 50 Hz downfield from the α-ATP signal. Chemical shifts in all ^31^P metabolites were referenced to PCr at 0 ppm. Gaussian apodization was applied to each free induction decay (FID) before Fourier transformation using the scanner software (SpectroView R5.7; Philips Healthcare).

### 2.3. ^31^P MRS Data Analysis

Frequency-domain ^31^P spectra were baseline-corrected, and signal intensities for Pi, PCr, and α-, β-, and γ-ATP were measured by integral (peak area under curve) and normalized by the corresponding signals in the reference spectrum to derive relative magnetizations. To evaluate ATP synthesis kinetic exchange parameters k_Pi→γATP_ and k_PCr→γATP_, the magnetizations were fit to a five-pool kinetic model using a home-built data analysis program written in Matlab (version R2021b, The MathWorks, Inc., Natick, MA, USA), as described previously [18].

The dynamic time constant τ_PCr_, an index of skeletal muscle mitochondrial oxidative phosphorylation, was evaluated by fitting the post-exercise PCr signals to the following mono-exponential equation:(1)$$\mathrm{PCr}(t)\;=\;\mathrm{PCr}{|}_{\infty }\;-\;\Delta \mathrm{PCr}\text{·}e^{-t/\tau_{PCr}}$$
where PCr|_∞_ is the [PCr] after full recovery, ΔPCr = PCr|_∞_ − PCr|_ex_, in which PCr|_ex_, is the end-of-exercise [PCr]. The initial rate of post-exercise PCr recovery V_PCr_ was evaluated by (1 − PCr|_ex_/PCr|_rest_,) [PCr]_o_/τ_PCr_ in which PCr|_rest_ represents the steady state [PCr] acquired from dynamic ^31^P spectra in resting muscle and [PCr]_o_ represents resting state [PCr] acquired from fully relaxed ^31^P spectra (see below).

Similarly, the decay time constant τ_Pi_ for the post-exercise recovery of accumulated Pi, which occurs through rephosphorylating ADP into ATP with coupled PCr replenishing (a process mediated by CK), was evaluated by fitting the post-exercise Pi signals to the following equation:(2)$$\mathrm{Pi}(t)\;=\;\mathrm{Pi}{|}_{\infty }\;+\;\Delta \mathrm{Pi}\cdot e^{-t/\tau_{Pi}}$$
where Pi|_∞_ is the [Pi] after full recovery, ΔPi = Pi|_ex_ − Pi|_∞_, in which Pi|_ex_ is the end-of-exercise [Pi]. The relative Pi accumulation at the end of exercise was evaluated by the ratio Pi|_∞_/Pi|_rest_, in which Pi|_rest_ represents the resting state [Pi]. The concentrations of Pi and PCr in resting muscle were evaluated using the fully relaxed ^31^P spectra acquired at TR = 30 s, with γ-ATP as the internal reference (set to 8.2 mM).

The pH was evaluated by the chemical shift of Pi signal in reference to PCr (set to 0 ppm) using the following formula:(3)$$\mathrm{pH}\;=\;\mathrm{pKa}\;+\;log\frac{\delta_{Pi}-\delta_{a}}{\delta_{b}-\delta_{Pi}}$$
where the H_2_PO_4_^−^ ↔ H^+^ + HPO_4_^2−^ deprotonation constant pKa = 6.73, and the ^31^P limiting shifts δ_a_ = 3.275 ppm (for acidic protonated species H_2_PO_4_^−^) and δ_b_ = 5.685 ppm (for basic deprotonated species HPO_4_^2−^) were used in the data analysis.

### 2.4. Statistical Analysis

All data were presented as mean ± standard deviation. Statistical significance (*p* < 0.05) for differences between two measurements—comparing the PD and control groups, as well as pre- and post-exercise training—was assessed using a *t*-test (at the 95% confidence level). The *t*-test was performed using the internal ttest2 function in MATLAB.

## 3. Results

### 3.1. Dynamic ^31^P MRS

Figure 1 shows the time series of representative dynamic ^31^P MR spectra acquired from calf muscle, along with the experimental setup of the RF coil and ergometer on the scanner table (insert, left panel). Distinct patterns of metabolite change were evident among PCr, Pi, and γ-ATP during 2 min of plantarflexion and post-exercise recovery (Figure 2A). Specifically, PCr was depleted while Pi accumulated during muscle contraction. Opposite trend toward recovery occurred during the subsequent muscle relaxing period, with both PCr and Pi characterized by a mono-exponential process (Figure 2B). In contrast, ATP signals remained nearly invariant in the entire time-course (Figure 2A, top). After Bo correction (set PCr signal as 0 ppm), all steady-state dynamic ATP signals agreed well in line-shape and width (Figure 2C), allowing highly repeatable data acquisition without Bo re-shimming (Figure 2C vs. Figure 2D).

Relative to the resting state, muscle PCr levels were depleted by 30–50% during this submaximal exercise, while Pi increased 2.5–3.5-fold at the cessation of exercise. However, no significant differences were found between PD and control groups in ΔPCr, ΔPi, or in the recovery time constants τ_PCr_ and τ_Pi_ (Table 1). The intracellular pH, derived from the chemical shift in the Pi signal, was slightly lower in PD compared to controls (pH 6.98 vs. pH 7.00, pre-exercise, Table 1). Upon exercise training, there was a minor increase in pH (by 0.03–0.04 units) for both groups. In contrast, the range of motion (ROM) for the plantarflexion weightlifting was significantly smaller in the PD group as compared to controls (6.0 ± 1.4 cm vs. 7.7 ± 1.3 cm, or 78 ± 18% *p* = 0.01, Table 1). ROM improved following exercise training in both groups; however, no statistically significant difference in ΔROM was observed between the PD and control groups (Table 1).

The post-exercise PCr recovery rate (V_PCr_) was correlated with the minutes of exercise (MOE) per week self-reported by the participants (*p* = 0.0011, Figure 3A), with individuals who exercise more routinely showing faster PCr recovery. A similar linear correlation was also observed between the dynamic time constants (τ_PCr_ and τ_Pi_) and MOE (Figure 3B,C).

### 3.2. Kinetic ^31^P MRS

In the kinetic ^31^P spectra acquired from resting calf muscle (Figure 4A–C), clear magnetization transfer (MT) effects due to chemical exchange were observed at Pi and PCr upon inversion of ATP signals, and at γ-ATP upon inversion of Pi and PCr. Quantitative Z-magnetization data analysis (Figure 4D–F) reveals that ATP synthesis, including CK- and ATPase-mediated reactions, was the main contributor to the MT effects at Pi, PCr, and γ-ATP. Without such MT effects, distinctly different magnetization trajectories would be predicted by those three inversion modules (Figure 4G–I, and Appendix A). It was found that ATP synthesis was significantly lower in PD compared to controls (*p* < 0.01, Figure 5). Specifically, k_PCr_ ranged from 0.18 to 0.28 s^−1^ in PD and from 0.25 to 0.35 s^−1^ in controls, while k_Pi_ was 5 to 7 times smaller in PD (Table 1, Figure 5A,B). Figure 5C compares ATP synthesis (k_Pi_ and k_PCr_) in resting muscle and the range of motion (ROM) observed during muscle exercise between the PD and control groups. Exercise training led to a slight increase in ATP synthesis kinetics in resting muscle for both groups, though the change did not reach statistical significance (Table 1). In addition, significant differences were observed in the ATP intramolecular ^31^P–^31^P cross-relaxation rates and the intrinsic T_1_ relaxation times of ATP (Appendix A). All these ^31^P NMR relaxation parameters remained largely unchanged following exercise training.

### 3.3. Correlation Between Kinetic and Dynamic Measurements

There was a weak correlation between the initial PCr recovery rate (V_PCr_) in post-exercise relaxing muscle and the ATP synthesis rate constants (k_Pi_ and k_PCr_, Figure 6A,B) and fluxes (F_Pi_ and F_PCr_, Figure 6C,D) in resting muscle. Specifically, faster ATP synthesis kinetics in resting muscle were associated with more rapid PCr recovery following exercise. The derived ratio V_PCr_/F_Pi_ was higher in the PD group compared to controls.

## 4. Discussion

### 4.1. Declined ATP Kinetic Activities in Resting Muscle

Assuming adequate blood flow and oxygen delivery in resting skeletal muscle, the observed ATP kinetics can be interpreted as reflective of mitochondrial oxidative function. Therefore, the reduced ATP kinetic activity observed in individuals with PD may suggest impaired mitochondrial function in skeletal muscle. Prior research has found mitochondrial dysfunction in the substantia nigra impacting PD disease pathogenesis [19]. Although limited, an increasing number of studies have also highlighted impairments in mitochondrial function within peripheral skeletal muscle [20,21,22]. Intuitively, reduced ATP synthesis kinetics may reflect a weaken skeletal muscle unable to meet acute energy demands, thereby hindering its transition from rest to movement. Such an energy mechanism may help explain the difficulty patients with PD often experience when initiating movement and slowness during movement. This metabolic-centric view is consistent with a recent study that found the metabolic defects in the muscles of PwP, suggesting a potential role for skeletal muscle intrinsic pathology in PD-related muscle dysfunction [21]. Declined ATP kinetic activities may also contribute to muscle freezing, a phenomenon often seen in the middle to late stages of PD, where patients experience a sudden and temporary loss of movement. However, the pathophysiology of freezing of gait is multimodal and remains unclear for any definitive mechanism at this time, with prior literature suggesting contributions from impairments in basal ganglia and central gait control mechanisms to cognitive and perceptual dysfunction [24]. In summary, the decline in resting muscle ATP kinetic activities may play a key role in the development and progression of major PD symptoms.

### 4.2. Reduced Range of Motion (ROM)

The observation of a significantly smaller ROM during plantarflexion in the PD group (78 ± 18% of controls) is consistent with the characteristic muscle rigidity and weakness in PD. Notably, despite this ROM limitation, the PD and control groups exhibited similar metabolic responses in pH, PCr, Pi, and post-exercise recovery rates (Table 1). Taken together, these findings suggest that muscle function in PD is metabolically less efficient, producing reduced movement or mechanical output despite comparable metabolic responses.

### 4.3. Increased PCr Reserve

Since the PCr pool in skeletal muscle serves as a readily available energy reserve to meet acute energy demands during exercise, the observed PCr increase in PD as compared to the control group may reflect an adaptive compensation mechanism to offset the reduced ATP kinetic activities in resting muscle in PD. Notably, a significant elevation in cellular PCr (*p* < 0.0001) has also been observed across multiple brain regions in patients with Pelizaeus–Merzbacher disease [25], a neurological disorder characterized by movement symptoms similar to those of PD, including muscle stiffness (spasticity), difficulty with movement and balance (ataxia), and tremors (though predominantly affecting the head and neck during movement) [26]. Phosphocreatine (PCr) and creatine (Cr) have been recognized as potent antioxidants with demonstrated neuroprotective properties in the brain [27], but whether these roles extend to skeletal muscle remains under investigation.

### 4.4. ATP Synthesis in Resting Versus Post-Exercise Muscle

Given that mitochondria’s primary function is to maintain ATP energy homeostasis, it is no surprise that the kinetic ATP flux in resting muscle correlates with the dynamic PCr recovery rate (Figure 6C), a measurement of mitochondrial oxidative phosphorylation capacity in post-exercise muscle. In this study, with 2 min rhythmic plantar-flexion exercise, the derived recovery time constants τ_PCr_ (Table 1) agree reasonably well with a previous PD study (PD: 59.6 ± 25.4 s, senior control: 44.5 ± 14.3 s, pre-training [23]). In addition, the mEBIT-derived de novo ATP synthesis rate constants and fluxes in controls (Table 1) are consistent with the early findings in the literature [28,29,30,31].

Although the term mitochondrial dysfunction is often used to describe impaired ATP energetics, reduced ATP flux may also result from unfavorable cellular environmental factors such as limited tissue oxygenation. Brief light-to-moderate exercise is commonly used as a warm-up to enhance muscle oxygenation; therefore, the observation of a V_PCr_/F_Pi_ greater than 1 (Table 1) may reflect a more favorable tissue oxygenation state in post-exercise muscle compared to resting muscle. The finding that the PD group exhibited a higher V_PCr_/F_Pi_ ratio than the control group may suggest compromised resting muscle oxygenation in PwP.

Additional support for this oxygenation hypothesis comes from the observation of longer ATP ^31^P T_1_ relaxation times in the resting muscle of PwP compared to controls (Appendix A). Since oxygen (O_2_) is paramagnetic, reduced tissue O_2_ perfusion is expected to prolong T_1_ relaxation times. This also helps explain why the brain—one of the most highly oxygenated and metabolically active organs—exhibits much shorter T_1_ times than resting skeletal muscle [18]. Furthermore, reduced resting muscle oxygenation in PD may reflect an adaptive response to decreased physical activity and lower total daily exercise volume in PwP. In addition to the oxygenation factor, other chronic conditions—such as obesity, heart failure, and increased dietary phosphate intake have also been shown to negatively affect mitochondrial function and ATP synthesis [32,33,34].

### 4.5. Limitations

This study has several limitations. First, using a non-localized ^31^P-MRS sequence means that the metabolite ^31^P signals obtained include contributions from both gastrocnemius and soleus muscles. Although the gastrocnemius lies in the RF coil’s more sensitive zone and we instructed participants to perform plantarflexion with a straight knee, plus using a knee strap to minimize potential knee binding during exercise and maximally engage the gastrocnemius, cross-muscle contamination cannot be ruled out. Future work should employ localized ^31^P-MRS techniques—such as sLASER single voxel, ISIS, or DRESS sequences—that enable selective interrogation of individual muscles and minimize signal contamination. Ideally, these sequences should also be compatible with kinetic MRS (e.g., mEBIT) and support acquisition at high temporal resolution for use in in-magnet exercise paradigms [35].

Second, the in-magnet exercise protocol used in this study involved 2 min of plantar flexion with a 2 s repetition cycle at a workload of 15% lean-body mass. Further improvements on this exercise protocol may be necessary to better mimic clinical exercise training routines and induce more pronounced metabolic changes in pH, PCr, and Pi, which could help differentiate patients with PD from controls.

Finally, the exercise intervention in this protocol was relatively short, lasting only 4 weeks, and involved a small sample size. As such, the findings from this pilot study should be validated through further studies with larger cohorts and more comprehensive exercise interventions. Implementing an endurance–strength combination protocol may induce larger and more sustainable beneficial effects, and better suited for validation by metabolic imaging. Indeed, it has been noted that a 3-month endurance–strength training improves metabolism and clinical state in PwP [23], and that prescription of high-intensity exercise to PwP improves mitochondrial function, muscle mass and physical capacity [36]. An optimized exercise training program tailored to individual metabolic features can be integrated into comprehensive treatment plans for PwP. This could ultimately support more targeted, evidence-based rehabilitation strategies aimed at improving QoL and slowing PD disease progression.

### 4.6. Other Remarks

The results from this combined kinetic and dynamic ^31^P MRS study suggest that muscle improvements with strengthening training are primarily due to enhanced neural recruitment and neuromuscular engagement, potential improvement in peripheral muscle oxygenation and responsive metabolic adjustments. In terms of intervention strategies to improve QoL in PD, alongside exercise training to address muscle deconditioning and dopaminergic drug treatments to improve motor control, approaches targeting peripheral mitochondrial dysfunction and responsible underlying factors should also be considered. For instance, some patients with PD have reported beneficial effects from glutathione supplements and a ketogenic diet, both of which may enhance mitochondrial energy production by reducing reactive oxygen species (ROS) that can damage mitochondrial function [37,38,39]. While ameliorating neuronal mitochondrial dysfunction has long been a PD treatment target [40,41], it is equally important to emphasize the role of skeletal muscle in this context.

## 5. Conclusions

More than 200 years after Parkinson’s disease (PD) was first described, a cure remains elusive. Findings from this pilot study suggest that mitochondrial dysfunction may contribute to impaired skeletal muscle ATP metabolism in PD, leading to reduced peripheral strength and aggravation of motor symptoms. These results add to a growing body of evidence indicating that PD is not solely a neurodegenerative disorder, but a systemic metabolic syndrome marked by disrupted cellular energetics in both the central nervous system and peripheral tissues. Regular exercise appears to promote muscle health through adaptive metabolic responses. These insights highlight the importance of developing strategies that target both central motor control and peripheral muscle function in the comprehensive management of PD.

## Figures and Tables

**Figure 1 diagnostics-15-02573-f001:**
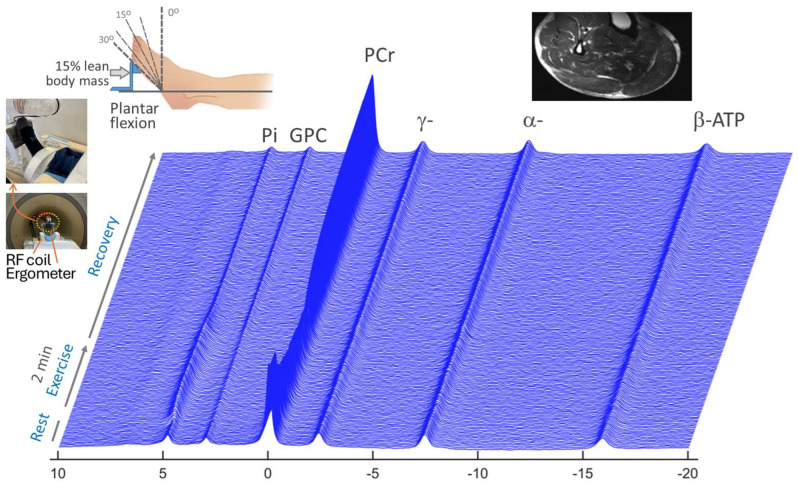
Dynamic 7T ^31^P MR spectra acquired from the calf muscle of patients with PD, showing PCr depletion concurrent with Pi accumulation during muscle contraction and their subsequent recoveries toward the resting state. The in-magnet exercise was a 2 min rhythmic plantar flexion at a repetition frequency of 2 s/cycle, pushing against a workload at 15% lean body mass.

**Figure 2 diagnostics-15-02573-f002:**
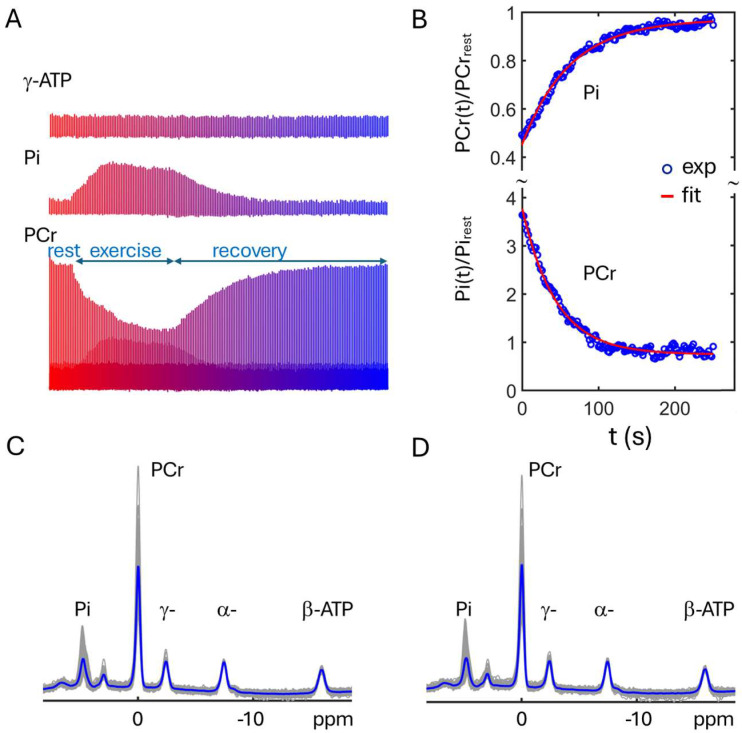
(**A**) Time course of PCr, Pi, and γ-ATP ^31^P signals, showing contrast in PCr depletion, Pi increase, while ATP remains homeostasis during exercise. (**B**) Post-exercise PCr and Pi recovery and curve fitting for evaluation of their recovery time constants. (**C**,**D**) Overlay of dynamic ^31^P spectra with Bo correction, showing reproducibility of experimental data in lineshape and width, insensitive to potential Bo change from exercise effects (**C** versus **D**).

**Figure 3 diagnostics-15-02573-f003:**
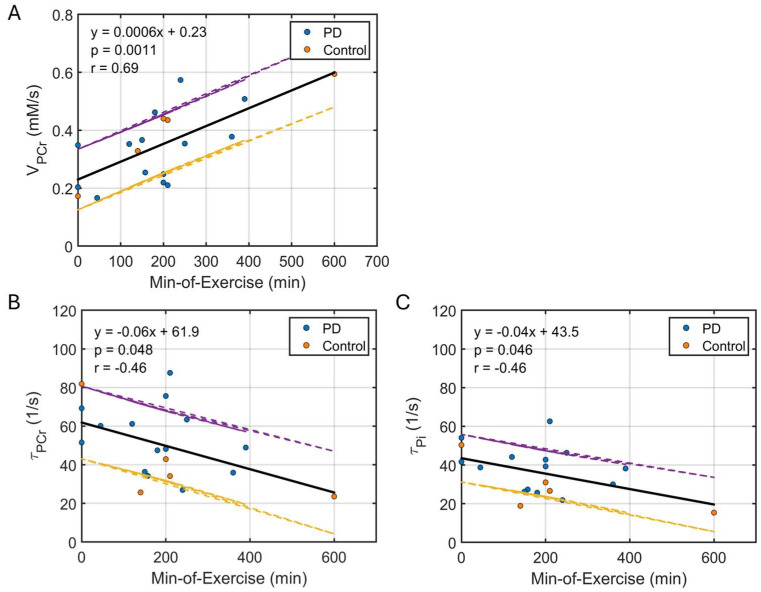
Plots of linear correlation between self-reported parameter min-of-exercise and dynamic-MRS measured parameters V_PCr_ (**A**), τ_PCr_ (**B**), and τ_Pi_ (**C**). The fitted regression line is shown in solid back, with the 95% uncertainty interval indicated by dashed yellow (lower bound) and purple (upper bound) lines.

**Figure 4 diagnostics-15-02573-f004:**
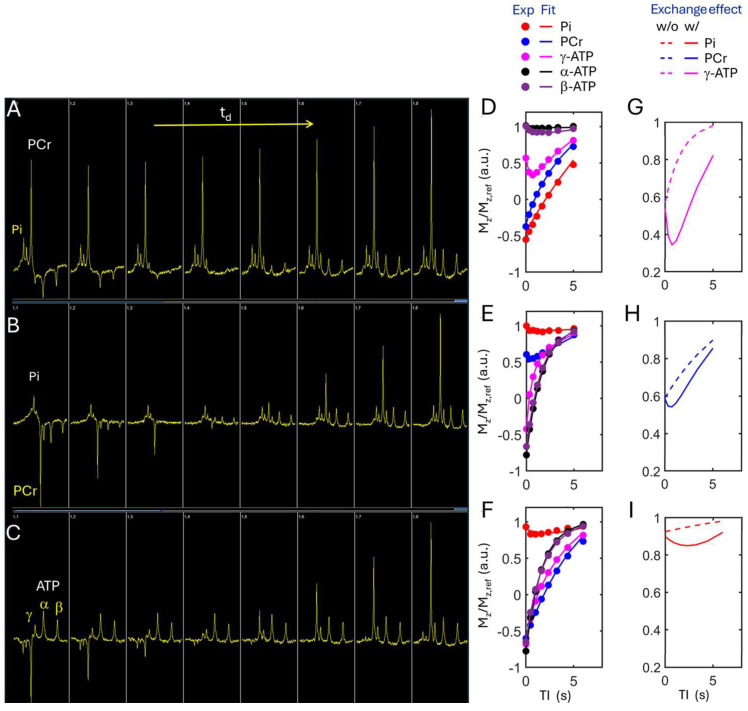
(**A**–**C**) Screenshots of kinetic ^31^P MR spectra acquired from calf muscle at rest by mEBIT sequence. (**D**–**F**) Experimental relative Z-magnetizations versus inversion-recovery time (TI) and their curve fitting according to the mEBIT method. (**G**–**I**) Comparison of relative Z-magnetization evolution with and without chemical exchange and ATP ^31^P-^31^P NOE effects for selected spins (for comparison of all 5 spins, see Appendix A). (**A**,**D**,**G**) mEBIT Module I, with inversion of α-, β-, and γ-ATP. Pi and PCr. (**B**,**E**,**H**) Module II, inversion of PCr and α-, β-, and γ-ATP. (**C**,**F**,**I**) Module III, with inversion of Pi and PCr.

**Figure 5 diagnostics-15-02573-f005:**
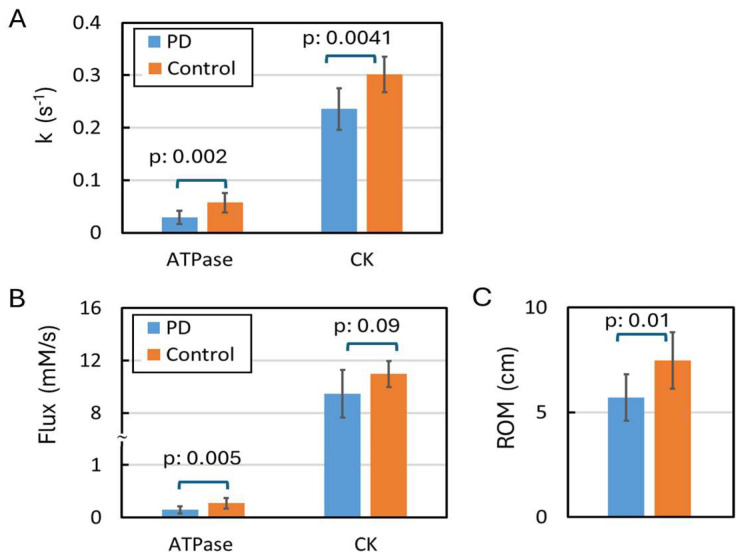
Comparison between the PD and control groups. (**A**) kinetic rate constants and (**B**) fluxes of ATP synthesis catalyzed by APTase and creatine kinase (CK) at the resting state. (**C**) Range of motion during in-magnet plantar flexion exercise.

**Figure 6 diagnostics-15-02573-f006:**
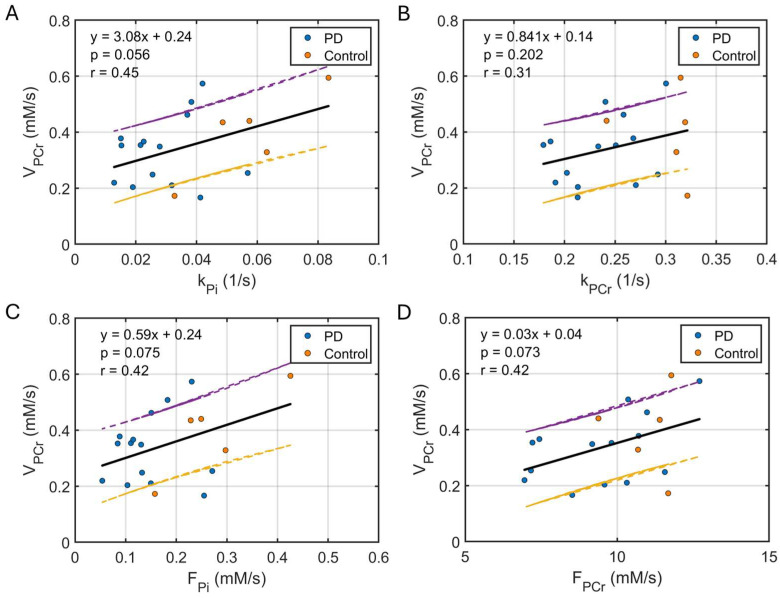
Plots of linear correlation between post-exercise PCr recovery rate (V_PCr_) and forward kinetic rate constants k_Pi_ (**A**) and k_PCr_ (**B**) and fluxes F_Pi_ (**C**) and F_PCr_ (**D**) for ATP synthesis in resting muscle. The fitted regression line is shown in solid back, with the 95% uncertainty interval indicated by dashed yellow (lower bound) and purple (upper bound) lines.

**Table 1 diagnostics-15-02573-t001:** Results of kinetic and dynamic ^31^P MRS data analysis.

	PD		Control			*p*-Val	
	Pre	Post	Pre	Post	PD ^1^	Ctrl ^1^	PD-vs-Ctrl ^2^
k_Pi_, s^−1^	0.03	0.03	0.06	0.05	0.87	0.39	0.0016 *
	(0.02)	(0.01)	(0.02)	(0.02)			
k_PCr_, s^−1^	0.23	0.24	0.32	0.29	0.37	0.25	0.0041 *
	(0.05)	(0.04)	(0.04)	(0.04)			
F_Pi_, mM/s	0.15	0.15	0.31	0.24	0.93	0.39	0.0054 *
	(0.09)	(0.07)	(0.13)	(0.12)			
F_PCr_, mM/s	8.92	10.01	11.37	10.61	0.16	0.39	0.094
	(2.08)	(1.87)	(1.36)	(1.31)			
Pi, mM	5.09	5.11	4.85	4.26	0.94	0.42	0.078
	(0.55)	(0.97)	(0.80)	(1.32)			
PCr, mM	38.9	41.1	35.9	37.2	0.05	0.30	0.012 *
	(2.2)	(3.2)	(2.2)	(1.5)			
τ_PCr_, s^−1^	51.5	55.2	40.1	43.2	0.60	0.84	0.25
	(18.7)	(14.8)	(22.0)	(25.5)			
τ_Pi_, s^−1^	38.6	38.4	23.2	33.7	0.98	0.31	0.13
	(11.8)	(13.4)	(7.6)	(20.2)			
V_PCr_, mM/s	0.31	0.35	0.37	0.42	0.40	0.61	0.37
	(0.13)	(0.13)	(0.12)	(0.20)			
V_PCr_/F_Pi_, a.u.	2.73	3.07	1.39	2.22	0.62	0.25	0.09
	(1.48)	(2.02)	(0.83)	(1.24)			
PCr_ex_/PCr_rest_	0.62	0.66	0.64	0.60	0.23	0.50	0.65
	(0.14)	(0.13)	(0.09)	(0.09)			
Pi_ex_/Pi_rest_	2.99	2.82	2.69	3.22	0.08	0.30	0.39
	(0.92)	(0.79)	(0.88)	(0.60)			
pH_rest_	7.02	7.02	7.04	7.05	0.67	0.67	0.13
	(0.04)	(0.04)	(0.03)	(0.03)			
pH_ex_	6.98	7.01	7.00	7.04	0.32	0.21	0.16
	(0.09)	0.06	(0.05)	(0.04)			
ROM ^3^, cm	5.90	6.12	7.10	8.31	0.20	0.20	0.01 *
	(1.52)	(1.36)	(1.43)	(1.07)			

^1^ *t*-test for differences in measurement between pre- and post-exercise training. ^2^ *t*-test for differences in measurement between PD and controls, with measurements averaged over pre- and post-rehab. ^3^ ROM: range of motion. * indicates statistical significance at *p* < 0.05.

## Data Availability

Data in this study are available upon request to the corresponding authors. The data are not publicly available due to privacy and legal reasons.

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
