# Peer review of "Skeletal Muscle 31P Magnetic Resonance Spectroscopy Study of Patients with Parkinson’s Disease: Energy Metabolism and Exercise Performance"

_diagnostics, 2025, doi:10.3390/diagnostics15202573_

Round 1

Reviewer 1 Report

Comments and Suggestions for Authors

This is a nice and well-written manuscript. I have a few minor comments. 

  • Include information (maybe pictures) about the RF coil (i.e. diameter, double-tuned, T/Rx? ...) and ergometer (for both legs or only one leg, manufacturer). 
  • Why did the authors set it to 15% of body weight and 30 degree?
  • How did you set the saturation frequency for gamma-ATP (doublet) and Pi (in-/ex-)? And for the quantification?
  • Any comments about the disturbance of static B0 shimming resulted by the exercise?
  • There are similar studies looking at the effect and it would be valuable to include a short discussion regarding the correlation between the skeletal muscle and other body parts (e.g. heart).  

Reviewer 2 Report

Comments and Suggestions for Authors

This study evaluates skeletal muscle energy metabolism in PD. For this purpose, an innovative pilot study using 31P MRS was presented. The examination of muscle metabolism during both rest and exercise conditions highlights the paper's innovative nature and is important for revealing changes in ATP metabolism. The study demonstrates that PD is not only a central nervous system disease but also a systemic metabolic dysfunction. Furthermore, the conclusion that regular exercise can support peripheral metabolic adaptations is a clinically valuable finding. Although the paper is well-organized, several corrections would be important to improve its quality:
- The introduction section does not sufficiently address the shortcomings of existing studies. Therefore, a section should be included that highlights the points that make the paper stand out in the literature, compares it to existing studies, and explains its main contributions.
- The biggest problem is that the sample size used in the article is limited. How can we be sure that these values, 14 PD patients and 5 controls, are sufficient and the performance obtained is not limited in the generalizability of the results? Validation with larger patient and control groups is required.
- Demographic characteristics of patient groups, such as gender, age, and disease stage, should be provided in more detail. Results should be reported accordingly, and a discussion should be conducted regarding the potential effects of these characteristics on the results.
- This study focused solely on the gastrocnemius muscle for the muscle group. However, the evaluation of the approach on different muscle groups is also interesting. Therefore, studies on different muscles could have yielded holistic results.
- The formulas (Eq. 1-3) in section 2.3 are difficult to read. It would be more appropriate to give it in a text box.
- In the 3rd Result section, the evidence for the distinction between control and peripheral metabolic effects should be clarified, and the necessary evaluations should be made. A more comprehensive and in-depth analysis of Figures 2, 3, and 4 would be appropriate.
- It can be said that the exercise protocol is limited. (planta flexion, calf-raise) Therefore, could more details have been included, such as more comprehensive intensity, duration and individual adaptation differences?
- Further discussion of the study's potential for translation into clinical practice could be included in the 5.Conclusions section, particularly in areas such as exercise prescriptions or integration into treatment plans.

Round 2

Reviewer 2 Report

Comments and Suggestions for Authors

The authors appear to have fully completed the necessary revisions to the article and have made significant improvements. I believe the paper does not need further revision.